# Robustness of Model Predictions under Extension

**Tineke Blom**[1]                                    **Joris M. Mooij**[2]

[1]Informatics Institute, University of Amsterdam, The Netherlands
[2]Korteweg-De Vries Institute for Mathematics, University of Amsterdam, The Netherlands

## Abstract

Mathematical models of the real world are simplified representations of complex systems. A caveat to using mathematical models is that predicted causal effects and conditional independences may not be robust under model extensions, limiting applicability of such models. In this work, we consider conditions under which qualitative model predictions are preserved when two models are combined. Under mild assumptions, we show how to use the technique of *causal ordering* to efficiently assess the robustness of qualitative model predictions. We also characterize a large class of model extensions that preserve qualitative model predictions. For dynamical systems at equilibrium, we demonstrate how novel insights help to select appropriate model extensions and to reason about the presence of feedback loops. We illustrate our ideas with a viral infection model with immune responses.

## 1 INTRODUCTION

A popular class of mathematical models that can represent uncertainty and causality are Structural Causal Models (SCMs) [Pearl, 2009, Bongers et al., 2021]. However, there are several interesting systems for which the causal relations and Markov properties cannot be modelled by SCMs [Blom et al., 2019]. The causal ordering algorithm, introduced by Simon [1953], can be used to deduce the qualitative predictions of mathematical models for these systems with regards to the causal relations between the variables in the system, and the probabilistic independence relations between the variables [Blom et al., 2021]. In this paper, we take a closer look at what happens to these predictions when two systems are combined. Particularly, we give conditions under which properties of the whole system can be understood in terms

of properties of its parts. We discuss how a holistic approach towards causal modelling may result in novel insights when we derive and test the predictions of systems for which new properties emerge from the combination of its parts.

In the first part of the paper, we focus on the practical issue of assessing whether qualitative model predictions are robust under model extensions. We revisit the observations of De Boer [2012] who demonstrated that qualitative predictions of a certain viral infection model change dramatically when the model is extended with extra equations describing simple immune responses. To assess the robustness of predicted causal relations or conditional independences under such an alteration of the model, it is useful to characterize a class of model extensions that lead to unaltered qualitative model predictions. In this work, we propose the technique of causal ordering [Simon, 1953] as an efficient method to assess the robustness of qualitative causal predictions. Under mild conditions, this allows us to characterize a large class of model extensions that preserve qualitative causal predictions. We also consider the class of models that are obtained from the equilibrium equations of dynamical models where each variable is *self-regulating*. For this class, we show that the predicted presence of causal relations and absence of conditional independences is robust when the model is extended with new equations.

Key aspects of the scientific method include generating a model or hypothesis that explains a phenomenon, deriving testable predictions from this model or hypothesis, and designing an experiment to test these predictions in the real world. The promise of causal discovery algorithms is that they are able to learn causal relations from a combination of background knowledge and data. The general idea of many constraint-based approaches (e.g. PC or FCI and variants thereof [Spirtes et al., 2000, Zhang, 2008, Colombo et al., 2012]) is to exploit information about conditional independences in a probability distribution to construct an equivalence class of graphs that encode certain aspects of the probability distribution, and then draw conclusions about the causal relations from the graphs. There is a large

*Accepted for the 38th Conference on Uncertainty in Artificial Intelligence* (UAI 2022).

amount of literature concerning particular algorithms for which the learned structure expresses causal relations under various combinations of assumptions (e.g. linearity, causal sufficiency, absence of feedback loops), see for example [Richardson and Spirtes, 1999, Spirtes et al., 2000, Lacerda et al., 2008, Zhang, 2008, Colombo et al., 2012, Hyttinen et al., 2012, Forré and Mooij, 2018, Strobl, 2019, Mooij and Claassen, 2020]. In the last part of this paper, our main interest is in dynamical models with the property that directed graphs representing relations between variables by encoding the conditional independences of their equilibrium distribution should not be interpreted causally at all. For the case that a model for a subsystem is given, we present novel insights that enable us to reject model extensions based on conditional independences in equilibrium data of the subsystem. We demonstrate how this approach allows us to reason about the presence of variables that are not self-regulating and feedback mechanisms that involve unobserved variables from the equilibrium distribution of certain dynamical models.

## 1.1 CAUSAL ORDERING GRAPH

Here, we give a concise introduction to the technique of causal ordering, introduced by Simon [1952].[1] In short, the causal ordering algorithm takes a set of equations as input and returns a *causal ordering graph* that encodes the effects of interventions and a *Markov ordering graph* that implies conditional independences between variables in the model [Theorem 17, Blom et al., 2021]. Compared with the popular framework of structural causal models [Pearl, 2009, Bongers et al., 2021], the distinction between the causal ordering and Markov ordering graphs does not provide new insights for acyclic models, but it results in nontrivial conclusions for models with feedback, as suggested in the discussion in Section 2.4 and explained in detail in [Blom et al., 2021].

We consider models consisting of equations $F$ that contain endogenous variables $V$, independent exogenous random variables $W$, and (constant, exogenous) parameters $P$. The structure of equations and the endogenous variables that appear in them can be represented by the *associated bipartite graph* $\mathcal{B} = \langle V, F, E \rangle$, where each endogenous variable is associated with a distinct vertex in $V$, and each equation is associated with a distinct vertex in $F$. There is an edge $(v - f) \in E$ if and only if variable $v \in V$ appears in equation $f \in F$. The causal ordering algorithm constructs a *directed cluster graph* $\langle \mathcal{V}, \mathcal{E} \rangle$, where $\mathcal{V}$ is a partition of vertices $V \cup F$ into clusters and $\mathcal{E}$ is a set of directed edges from vertices in $V$ to clusters in $\mathcal{V}$. Given a bipartite graph

$\mathcal{B} = \langle V, F, E \rangle$ with a perfect matching[2] $M$, the causal ordering algorithm proceeds with the following three steps [Nayak, 1995, Blom et al., 2021]:

1. For $v \in V$, $f \in F$ orient edges $(v - f)$ as $(v \leftarrow f)$ when $(v - f) \in M$ and as $(v \rightarrow f)$ otherwise; this yields a directed graph $\mathcal{G}(\mathcal{B}, M)$.

2. Find all strongly connected components $S_1, S_2, \ldots, S_n$ of $\mathcal{G}(\mathcal{B}, M)$. Let $\mathcal{V}$ be the set of clusters $S_i \cup M(S_i)$ for $i \in \{1, \ldots, n\}$, where $M(S_i)$ denotes the set of vertices that are matched to vertices in $S_i$ in matching $M$.

3. Let $\mathrm{cl}(f)$ denote the cluster in $\mathcal{V}$ containing $f$. For each $(v \rightarrow f)$ in $\mathcal{G}(\mathcal{B}, M)$ such that $v \notin \mathrm{cl}(f)$ add an edge $(v \rightarrow \mathrm{cl}(f))$ to $\mathcal{E}$.

Independent exogenous random variables and parameters are then added as singleton clusters with edges towards the clusters of the equations in which they appear. It has been shown that the resulting directed cluster graph $\mathrm{CO}(\mathcal{B}) = \langle \mathcal{V}, \mathcal{E} \rangle$, which we refer to as the *causal ordering graph*, is independent of the choice of perfect matching [Theorem 4, Blom et al., 2021]. Example 1 shows how the algorithm works and a graphical illustration of the algorithm for a more elaborate cyclic model can be found in Section A of the Supplementary Material.

**Example 1.** Let $V = \{v_1, v_2\}$, $W = \{w_1, w_2\}$, and $P = \{p_1, p_2\}$ be index sets. Consider model equations $f_1$ and $f_2$ with endogenous variables $(X_v)_{v \in V}$, exogenous random variables $(U_w)_{w \in W}$ and parameters $(C_p)_{p \in P}$:

$$f_1: \quad C_{p_1} X_{v_1} - U_{w_1} = 0, \tag{1}$$
$$f_2: \quad C_{p_2} X_{v_2} + X_{v_1} + U_{w_2} = 0. \tag{2}$$

The bipartite graph $\mathcal{B} = \langle V, F, E \rangle$ in Figure 1a, with $E = \{(v_1 - f_1), (v_1 - f_2), (v_2 - f_2)\}$ is a compact representation of the model structure. This graph has a perfect matching $M = \{(v_1 - f_1), (v_2 - f_2)\}$. By orienting edges in $\mathcal{B}$ according to the rules in step 1 of the causal ordering algorithm we obtain the directed graph $\langle V \cup F, E_{\mathrm{dir}} \rangle$ with $E_{\mathrm{dir}} = \{(f_1 \rightarrow v_1), (f_2 \rightarrow v_2), (v_1 \rightarrow f_2)\}$. The clusters $C_1 = \{v_1, f_1\}$ and $C_2 = \{v_2, f_2\}$ are added to $\mathcal{V}$ in step 2 of the algorithm, and the edge $(v_1 \rightarrow C_2)$ is added to $\mathcal{E}$ in step 3. Finally, we add the parameters $P$ and independent exogenous random variables $W$ as singleton clusters to $\mathcal{V}$, and the edges $(p_1 \rightarrow C_1)$, $(w_1 \rightarrow C_1)$, $(p_2 \rightarrow C_2)$, and $(w_2 \rightarrow C_2)$ to $\mathcal{E}$. The resulting causal ordering graph is given in Figure 1b. $\triangle$

---

[1] Actually, we consider an equivalent algorithm for causal ordering that was shown to be more computationally efficient by [Nayak, 1995, Gonçalves and Porto, 2016]. For more details, see [Blom et al., 2021].

[2] A perfect matching $M$ is a subset of edges in a bipartite graph so that every vertex is adjacent to exactly one edge in $M$. If $(v - f) \in M$ we say that $v$ and $f$ are matched in $M$. Note that not every bipartite graph has a perfect matching; the theory can be extended to bipartite graphs that have no perfect matching by making use of maximal matchings instead [Blom et al., 2021].

Throughout this work, we will assume that models are *uniquely solvable with respect to the causal ordering graph*, which roughly means that for each cluster, the equations in that cluster can be solved uniquely for the endogenous variables in that cluster (see [Definition 14, Blom et al., 2021] for details). A *perfect intervention on a cluster* that contains equation vertices represents a model change where the equations in the targeted cluster are replaced by equations that set the endogenous variables in that cluster equal to constant values. A *soft intervention* targets an equation, parameter, or exogenous variable, but does not affect which variables appear in the equations. We say that there is a *directed path* from a vertex $x$ to a vertex $y$ in a causal ordering graph $\langle \mathcal{V}, \mathcal{E} \rangle$ if either $\mathrm{cl}(x) = \mathrm{cl}(y)$ or there is a sequence of clusters $C_1 = \mathrm{cl}(x), C_2, \ldots, C_{k-1}, C_k = \mathrm{cl}(y)$ so that for all $i \in \{1, \ldots, k-1\}$ there is a vertex $z_i \in C_i$ such that $(z_i \to C_{i+1}) \in \mathcal{E}$. It can be shown that (a) the presence of a directed path from a cluster, equation, parameter, or exogenous variable that is targeted by a soft intervention towards a certain variable in the causal ordering graph implies that the intervention has a generic effect on that variable, and (b) if no such path exists there is no causal effect of the intervention on that variable [Theorem 20, Blom et al., 2021]. For a perfect intervention that targets a cluster in the causal ordering graph, one can similarly read off its non-effects and generic effects from the causal ordering graph [Theorem 23, Blom et al., 2021].

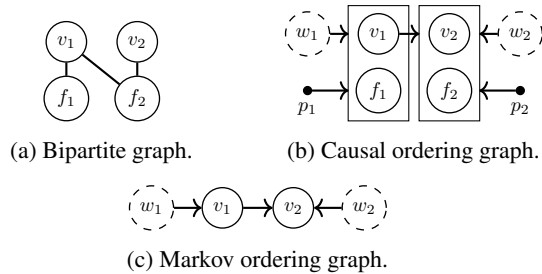

(a) Bipartite graph.  (b) Causal ordering graph.

(c) Markov ordering graph.

Figure 1: The bipartite graph in Figure 1a is a compact representation of the model in Example 1. The corresponding causal ordering graph and Markov ordering graph are given in Figures 1b and 1c respectively. Exogenous variables are denoted by dashed circles and parameters by black dots.

## 1.2 MARKOV ORDERING GRAPH

The causal ordering graph $\mathrm{CO}(\mathcal{B}) = \langle \mathcal{V}, \mathcal{E} \rangle$ of model equations $F$ with endogenous variables $V$, exogenous random variables $W$, parameters $P$, and bipartite graph $\mathcal{B}$ can be used to construct the *Markov ordering graph*, which is a DAG $\mathrm{MO}(\mathcal{B}) = \langle V \cup W, E \rangle$, with $(x \to y) \in E$ if and only if $(x \to \mathrm{cl}(y)) \in \mathcal{E}$. The Markov ordering graph for the model equations in Example 1 is given in Figure 1c. It has been shown that, under the assumption of unique solvability w.r.t. the causal ordering graph, d-separations in the Markov

ordering graph imply conditional independences between the corresponding variables [Blom et al., 2021]. Henceforth, we will assume that the probability distribution of the solution $(X_v)_{v \in V}$ to a set of model equations is faithful to the Markov ordering graph. In other words, each conditional independence in the distribution implies a d-separation in the Markov ordering graph. Under the assumption that data is generated from such a model, some causal discovery algorithms, such as the PC algorithm [Spirtes et al., 2000], aim to construct the Markov equivalence class of the Markov ordering graph. In this work, we will specifically focus on feedback models for which the Markov ordering graph of the equilibrium distribution, and consequently the output of many causal discovery algorithms, does not have a straightforward causal interpretation.

## 2 CAUSAL ORDERING FOR A VIRAL INFECTION MODEL

This work was inspired by a viral infection model discussed by De Boer [2012], who showed through explicit calculations that the predictions of the model are not robust under addition of an immune response. This sheds doubt on the correct interpretation of variables and parameters in the model. For many systems it is intrinsically difficult to study their behavior in detail. The use of simplified mathematical models that capture key characteristics aids in the analysis of certain properties of the system. The hope is that the explanations inferred from model equations are legitimate accounts of the true underlying system [De Boer, 2012]. In reality, a modeler must take into account that the outcome of these studies may be contingent on the specifics of the model design. Here, we demonstrate how causal ordering can be used as a scalable tool to assess the robustness of model predictions without requiring explicit calculations.

## 2.1 VIRAL INFECTION WITHOUT IMMUNE RESPONSE

Let $U_\sigma$ be a production term for target cells, $d_T$ the death rate for target cells, $U_f$ the fraction of successful infections, and $U_\delta$ the death rate of productively infected cells. Define $\beta = \frac{bp}{c}$, where $b$ is the infection rate, $p$ the amount of virus produced per infected cell, and $c$ the clearance rate of viral particles. The following first-order differential equations describe how the amount of target cells $X_T(t)$ and the amount of infected cells $X_I(t)$ evolve over time [De Boer, 2012]:

$$\dot{X}_T(t) = U_\sigma - d_T X_T(t) - \beta X_T(t) X_I(t), \quad (3)$$

$$\dot{X}_I(t) = (U_f \beta X_T(t) - U_\delta) X_I(t). \quad (4)$$

Suppose that we want to use this simple viral infection model to explain why the *set-point viral load* (i.e. the total amount of virus circulating in the bloodstream) of chronically infected HIV-patients differs by several orders of mag-

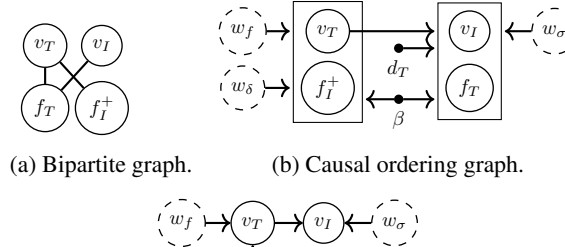

(a) Bipartite graph.      (b) Causal ordering graph.

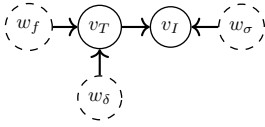

(c) Markov ordering graph.

Figure 2: Graphical representations of the viral infection model in equations (5) and (6). Vertices $v_i$ and $w_j$ correspond to variables $X_i$ and $U_j$, respectively. The causal ordering graph represents generic effects of interventions. The d-separations in Figure 2c imply conditional independences.

nitude, as De Boer [2012] does. To analyze this problem we look at the equilibrium equations that are implied by equations (3) and (4):[3]

$$f_T: \qquad U_\sigma - d_T X_T - \beta X_T X_I = 0, \qquad (5)$$
$$f_I^+: \qquad U_f \beta X_T - U_\delta = 0. \qquad (6)$$

Throughout the remainder of this work we will use this *natural labelling* of equilibrium equations, where the equation derived from the derivative $\dot{X}_i(t)$ is labelled $f_i$. For first-order differential equations that are written in canonical form, $\dot{X}_i(t) = g_i(X(t))$, the natural labelling always exists.

Suppose that $U_\sigma$, $U_f$ and $U_\delta$ are independent exogenous random variables taking values in $\mathbb{R}_{>0}$ and $d_T$, $\beta$ are strictly positive parameters. The associated bipartite graph, causal ordering graph, and Markov ordering graph are given in Figure 2. The causal ordering graph tells us that soft interventions targeting $U_\sigma$, $U_f$, $U_\delta$, $d_T$, or $\beta$ generically have an effect on the equilibrium distribution of the amount of infected cells $X_I$. From here on, we say that the causal ordering graph of a model predicts the *generic presence* or *absence* of causal effects. The Markov ordering graph shows that $v_T$ and $w_\sigma$ are d-separated. This implies that the amount of target cells $X_T$ should be independent of the production rate $U_\sigma$ when the system is at equilibrium. Henceforth, we will say that the Markov ordering graph predicts the *generic presence* or *absence* of conditional dependences.

## 2.2 VIRAL INFECTION WITH A SINGLE IMMUNE RESPONSE

The viral infection model in equations (3) and (4) can be extended with a simple immune response $X_E(t)$ by adding the following dynamic and static equations:

$$\dot{X}_E(t) = (U_a X_I(t) - d_E) X_E(t), \qquad (7)$$
$$X_\delta(t) = d_I + U_k X_E(t), \qquad (8)$$

where $U_a$ is an activation rate, $d_E$ and $d_I$ are turnover rates and $U_k$ is a mass-action killing rate [De Boer, 2012]. Note that the exogenous random variable $U_\delta$ is now treated as an endogenous variable $X_\delta(t)$ instead. We derive the following equilibrium equations, using the natural labelling for equation (7):[4]

$$f_E^+: \qquad U_a X_I - d_E = 0, \qquad (9)$$
$$f_\delta: \qquad X_\delta - d_I - U_k X_E = 0, \qquad (10)$$

Henceforth, we will call the addition of equations $F_+$ to $F$ a *model extension*. Notice that when two sets of equations are combined, there may exist variables that were exogenous in the submodel (i.e. the original model) but that are endogenous within the whole model (i.e. the extended model). Generally, equations $F_+$ may contain endogenous variables in $V$ and exogenous variables in $W$ but they may also contain additional endogenous variables $V_+$, additional exogenous variables $W_+$ and additional parameters $P_+$. Parameters and exogenous random variables that appear in equations $F$ can appear as endogenous variables in $V_+$ and in the extended model $F_{\text{ext}} = F \cup F_+$. In that case, these variables are no longer considered to be parameters or exogenous variables within the extended model.

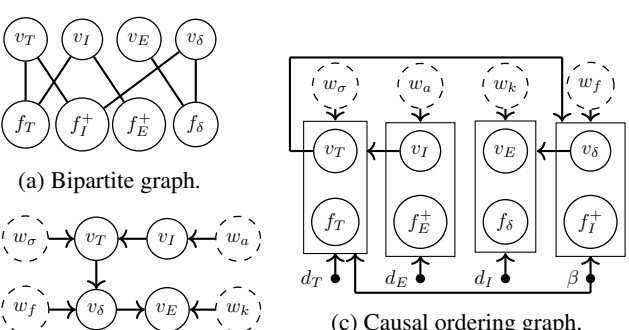

(a) Bipartite graph.

(b) Markov ordering graph.

(c) Causal ordering graph.

Figure 3: Graphical representations of the viral infection model with a single immune response. The presence or absence of causal relations and d-connections implied by the graphs in Figure 2 are not preserved if a single immune response is added.

Suppose that $U_a$ and $U_k$ are independent exogenous random variables taking values in $\mathbb{R}_{>0}$ and $d_E$, $d_I$ are parameters

---

[3]Since we are only interested in strictly positive solutions we removed $X_I$ from the equilibrium equation $f_I: (U_f \beta X_T - U_\delta) X_I = 0$ to obtain $f_I^+$.

[4]Analogous to changing $f_I$ to $f_I^+$ for strictly positive solutions, we will look at $f_E^+$ instead of $f_E$.

taking value in $\mathbb{R}_{>0}$. The bipartite graph, causal ordering graph, and Markov ordering graph associated with equations (5), (6), (9), and (10) (with $X_\delta$ replacing $U_\delta$) are given in Figure 3.

The causal ordering graph in Figure 3c predicts a causal effect of $U_\sigma$ and $d_T$ on $X_T$ but not on $X_I$. By comparing with the predictions of the causal ordering graph in Figure 2b (where we saw that soft interventions targeting $U_\sigma$ and $d_T$ generically do have an effect on $X_I$), we find that effects of interventions targeting $U_\sigma$ and $d_T$ are not robust under the model extension.

The Markov ordering graph of the extended model in Figure 3b shows that $w_\sigma$ is d-connected to $v_T$, and hence $U_\sigma$ and $X_T$ will be dependent at equilibrium for most parameter choices. On the other hand, in the Markov ordering graph of the viral infection model without immune response (Figure 2c), $w_\sigma$ and $v_T$ are d-separated, and hence, $U_\sigma$ and $X_T$ will be independent at equilibrium for any parameter choice according to the viral infection model without immune response. Therefore, the independence between $U_\sigma$ and $X_T$ is not robust under the model extension.

The systematic graphical procedure followed here easily leads to the same causal conclusions as De Boer [2012] obtained by explicitly solving the equilibrium equations. In addition, it leads to predictions regarding the conditional (in)dependences in the equilibrium distribution.

## 2.3 VIRAL INFECTION WITH MULTIPLE IMMUNE RESPONSES

The following static and dynamical equations describe multiple immune responses:

$$\dot{X}_{E_i}(t) = \frac{p_E X_{E_i}(t) U_{a_i} X_I(t)}{h + X_{E_i}(t) + U_{a_i} X_I(t)} - d_E X_{E_i}(t),$$
$$i = 1, 2, \ldots, n \tag{11}$$

$$X_\delta(t) = d_I + U_k \sum_{i=1}^{n} U_{a_i} X_{E_i}(t), \tag{12}$$

where there are $n$ immune responses, $U_{a_i}$ is the avidity of immune response $i$, $p_E$ is the maximum division rate, and $h$ is a saturation constant [De Boer, 2012]. For $n = 2$ we can derive equilibrium equations $f_{E_1}$, $f_{E_2}$, and $f_\delta$ using the natural labelling as we did for the equilibrium equations in the previous section. Together with the equilibrium equations (5) and (6) (with $X_\delta$ replacing $U_\delta$) for the viral infection model this is another extended model. The bipartite graph of this extended model is given in Figure 4a, while the causal ordering graph can be found in Figure 4b. By comparing the directed paths in this causal ordering graph with that of the original viral infection model (i.e. the model without an immune response) in Figure 2b, it can be seen that the predicted presence of causal relations is preserved under extension of the model with multiple immune responses, while

the predicted absence of causal relations is not. Similarly, by comparing d-separations in the Markov ordering graphs in Figure 2c with those in Figure 4c, we find that predicted conditional dependences are preserved under the extensions, while the predicted conditional independences are not.

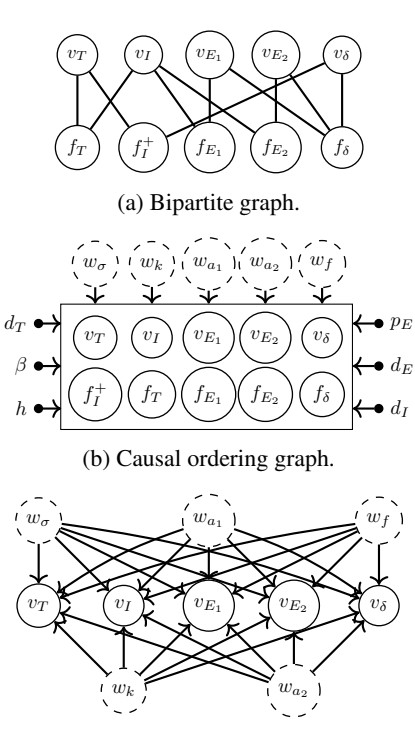

(a) Bipartite graph.

(b) Causal ordering graph.

(c) Markov ordering graph.

Figure 4: Graphical representations of the viral infection model with multiple immune responses. The presence of causal relations and d-connections in Figure 2 is preserved.

## 2.4 MARKOV ORDERING GRAPHS AND CAUSAL INTERPRETATIONS

In [Blom et al., 2021], it was shown that the Markov ordering graph may not have a straightforward causal interpretation. Here, we illustrate for the viral infection models that the Markov ordering graphs neither have a straightforward causal interpretation at equilibrium in terms of soft interventions targeting parameters or equations *nor* in terms of perfect interventions on variables in the dynamical model. To see this, consider the Markov ordering graph in Figure 3b for the viral infection with a single immune response. The edge $(v_I \rightarrow v_T)$ cannot correspond to the effect of a soft intervention targeting $f_I^+$, because the causal ordering graph in Figure 3c shows that there is no such effect. Clearly, directed paths in the Markov ordering graph do not necessarily represent the effects of soft interventions. A natural way to model a perfect intervention targeting a variable in the Markov ordering graph is to replace the (differential) equation of that variable with an equation setting that variable equal to a certain value in the underlying dy-

namical model [Mooij et al., 2013]. By explicitly solving equilibrium equations it is easy to check that replacing $f_E^+$ with an equation setting $X_E$ equal to a constant generically changes the equilibrium distributions of all four variables $X_E, X_\delta, X_T, X_I$. Since there are no directed paths from $v_E$ to any of $v_\delta, v_T, v_I$ in the Markov ordering graph in Figure 3b, the effects of this perfect intervention would not have been predicted by the Markov ordering graph, if it were interpreted causally. Therefore, contrary to the causal ordering graph, the Markov ordering graph does not have a causal interpretation in terms of soft or perfect interventions on the underlying dynamical model.

# 3 ROBUST CAUSAL PREDICTIONS UNDER MODEL EXTENSIONS

One way to gauge the robustness of model predictions is to check to what extent they depend on the model design. The example of a viral infection with different immune responses in the previous section indicates that qualitative causal predictions entailed by the causal ordering graph of a mathematical model may strongly depend on the particulars of the model. Both the implied presence or absence of causal relations at equilibrium and the implied presence or absence of conditional independences at equilibrium may change under certain model extensions. Under what conditions are these qualitative predictions preserved under model extensions? In this section, we characterize a large class of model extensions under which qualitative equilibrium predictions are preserved.

Theorem 1 gives a sufficient condition on model extensions under which the predicted generic presence of causal relations and predicted generic presence of conditional dependences at equilibrium is preserved. The proof is given in Section C of the Supplementary Material.

**Theorem 1.** *Consider model equations $F$ containing endogenous variables $V$ with bipartite graph $\mathcal{B}$. Suppose $F$ is extended with equations $F_+$ containing endogenous variables in $V \cup V_+$, where $V_+$ contains endogenous variables that are added by the model extension (which may include parameters or exogenous variables that appear in $F$ and become endogenous in the extended model). Let $\mathcal{B}_{\text{ext}}$ be the bipartite graph associated with $F_{\text{ext}} = F \cup F_+$ and $V_{\text{ext}} = V \cup V_+$, and $\mathcal{B}_+$ the bipartite graph associated with the extension $F_+$ and $V_+$, where variables in $V$ appearing in $F^+$ are treated as exogenous variables (i.e. they are not added as vertices in $\mathcal{B}_+$). If $\mathcal{B}$ and $\mathcal{B}_+$ both have a perfect matching then:*

1. *$\mathcal{B}_{\text{ext}}$ has a perfect matching,*
2. *ancestral relations in $\text{CO}(\mathcal{B})$ are also present in $\text{CO}(\mathcal{B}_{\text{ext}})$,*
3. *d-connections in $\text{MO}(\mathcal{B})$ are also present in $\text{MO}(\mathcal{B}_{\text{ext}})$.*

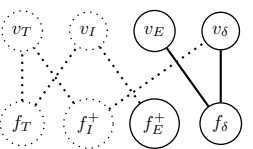 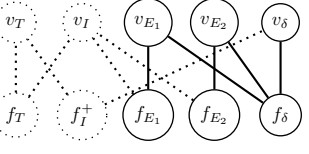

(a) Single response.  (b) Multiple responses.

Figure 5: The solid lines indicate the bipartite graphs $\mathcal{B}_+$ associated with the single immune response extension (5a) and the multiple immune response extension (5b). Combined with the dotted lines, one obtains the bipartite graphs $\mathcal{B}_{\text{ext}}$ for the complete models.

This result characterizes a large set of extensions under which the implied causal effects and conditional dependences of a model are preserved. Consider again the equilibrium behavior of the viral infection models in Section 2. We already showed explicitly that the extension of the viral infection model with multiple immune responses preserved the predicted presence of causal relations and conditional dependences, but with the help of Theorem 1 we only would have needed to check whether the bipartite graph in Figure 5b has a perfect matching to arrive at the same conclusion. The bipartite graph for the extension with a single immune response in Figure 5a does not have a perfect matching and hence the conditions of Theorem 1 do not hold. Recall that this model extension did not preserve the predicted presence of causal relations.

The theorem below gives a stronger condition under which (conditional) independence relations and the absence of causal relations that are implied by a model are also predicted by the extended model. The proof is provided in the supplement.

**Theorem 2.** *Let $F$, $F_+$, $F_{\text{ext}}$, $V$, $V_+$, $V_{\text{ext}}$, $\mathcal{B}$, $\mathcal{B}_+$, and $\mathcal{B}_{\text{ext}}$ be as in Theorem 1. If $\mathcal{B}$ and $\mathcal{B}_+$ both have perfect matchings and no vertex in $V_+$ is adjacent to a vertex in $F$ in $\mathcal{B}_{\text{ext}}$ then:*[5]

1. *ancestral relations absent in $\text{CO}(\mathcal{B})$ are also absent in $\text{CO}(\mathcal{B}_{\text{ext}})$,*
2. *d-connections absent in $\text{MO}(\mathcal{B})$ are also absent in $\text{MO}(\mathcal{B}_{\text{ext}})$.*

Together with Theorem 1, this result characterizes a large class of model extensions under which all qualitative model predictions are preserved. Consider again the equilibrium models for the viral infection in Section 2. The bipartite graph for the extension with a single immune response in Figure 5a does not have a perfect matching. In the bipartite graph associated with the viral infection model with

---

[5] A vertex in $V_+$ is considered adjacent to $F$ in $\mathcal{B}_{\text{ext}}$ if it corresponds with one of the exogenous random variables or parameters in $F$ that become endogenous in the model extension.

multiple immune responses in Figure 4a, the additional endogenous variable $v_\delta$ is adjacent to $f_I$. Neither of the model extensions satisfies the conditions of Theorem 2. We already demonstrated that neither of the model extensions preserves all qualitative model predictions. An example of a model extension that does satisfy the conditions in Theorem 1 and 2 is an acyclic structural causal model that is extended with another acyclic structural causal model such that the additional variables are non-ancestors of the original ones. Together, Theorem 1 and 2, can be used to understand when the causal and Markov properties of a composite system can be understood by studying the corresponding properties of its components.

# 4 SELECTION OF MODEL EXTENSIONS

So far, we have considered methods to assess the robustness of qualitative model predictions. In this section we will show how this idea results in novel opportunities regarding causal discovery. In particular, if we assume that the systems that we observe are part of a larger partially observed system, then we can use the methods in this paper to reason about causal mechanisms of unobserved variables. Consider, for example, the viral infection model for which we have demonstrated that extensions with different immune responses imply different (conditional) independences between variables in the original model. The Markov ordering graphs in Figures 2c, 3b, and 4c imply the following (in)dependences:

| Viral infection model | (In)dependences |
|---|---|
| without immune response | $U_\sigma \perp\!\!\!\perp X_T, U_\sigma \not\perp\!\!\!\perp X_I$ |
| with single immune response | $U_\sigma \not\perp\!\!\!\perp X_T, U_\sigma \perp\!\!\!\perp X_I$ |
| with multiple immune responses | $U_\sigma \not\perp\!\!\!\perp X_T, U_\sigma \not\perp\!\!\!\perp X_I$ |

Given a model for variables $X_T$ and $X_I$ only, we can reject certain model extensions based on the observed (conditional) independences for variables $X_T$, $X_I$, and $U_\sigma$ in data sampled from the distribution of the combined system—provided that we assume faithfulness and the correctness of the model of the original subsystem. Using this holistic modelling approach, we can reason about properties of an unknown model extension without observing the new mechanisms or variables. In the remainder of this section, we further discuss how this idea can be applied to equilibrium data of dynamical systems.

## 4.1 REASONING ABOUT SELF-REGULATING VARIABLES

We say that a variable in a set of first-order differential equations in canonical form is *self-regulating* if it can be solved uniquely from the equilibrium equation that is constructed from its derivative. For example, in the system of first-order ODEs (3–4), $X_T$ is self-regulating if $\beta X_I + d_T \neq 0$, whereas $X_I$ is not self-regulating.

For models in which every variable is self-regulating there exists a perfect matching where each variable $v_i$ is matched to its associated equilibrium equation $f_i$ according to the natural labelling, for more details see Lemma 1 in the supplement. Interestingly, the Markov ordering graph for the equilibrium equations of such a model always has a causal interpretation. By construction of the causal ordering graph from the bipartite graph and the perfect matching provided by the natural labelling, we know that a vertex $v_i$ always appears in a cluster with $f_i$ in the causal ordering graph. The presence or absence of directed paths in the Markov ordering graph can then easily be associated with the presence or absence of directed paths in the causal ordering graph. Consequently, the Markov ordering graph can be interpreted in terms of both soft interventions targeting equations and perfect interventions that set variables equal to a constant by replacement of the associated dynamical and equilibrium equations.[6]

For models in which every variable is self-regulating, it follows immediately from Theorem 1 that the presence of ancestral relations and d-connections is robust under dynamical model extensions in which each variable is self-regulating, as is stated more formally in Corollary 1 below.

**Corollary 1.** *Consider a first-order dynamical model in canonical form for endogenous variables $V$ and an extension consisting of canonical first-order differential equations for additional endogenous variables $V_+$. Let $F$ and $F_{\text{ext}} = F \cup F_+$ be the equilibrium equations of the original and extended model respectively. If all variables in $V \cup V_+$ are self-regulating, then statements 2 and 3 of Theorem 1 hold.*

Corollary 1 characterizes a class of models under which certain qualitative predictions for the equilibrium distribution are robust, but the result can also be interpreted from a different angle. Suppose that we have equilibrium data that is generated by an extended dynamical model with equilibrium equations $F_{\text{ext}}$, but we only have a *partial* model consisting of equations in $F$ for a subset $V \subseteq V_{\text{ext}} = V \cup V_+$ of variables that appear in $F_{\text{ext}} = F \cup F_+$. If we would find conditional independences between variables in $V$ that do not correspond to d-separations in the Markov ordering graph of the partial model, this does not necessarily mean that the model equations are wrong. It could also be the case, for example, that we are wrong to assume that the system can be studied in a reductionist manner and that the model should be extended. Furthermore, under the assumption that data is generated from the equilibrium distribution of a dynamical model, Corollary 1 tells us that conditional independences in the data that are not predicted by the equations

---

[6]Dynamical systems with only self-regulating variables were also considered in [Mooij et al., 2013], where it was shown that their equilibria can be modelled as structural causal models without self-cycles.

of a partial model imply the presence of variables that are not self-regulating, if we assume faithfulness. This shows that, given a model for a subsystem, we can reason about the properties of unobserved and unknown variables in the whole system. We will showcase an example for this type of reasoning in Section 4.3.

## 4.2 REASONING ABOUT FEEDBACK LOOPS

We say that an extension of a dynamical model *introduces a new feedback loop with the original dynamical model* when there is feedback in the extended dynamical model that involves variables in both the original model and the model extension. To make this definition more precise, consider the set $E_{\text{nat}}$ of edges $(v_i - f_i)$ that are associated with the natural labelling of the equilibrium equations of the extended dynamical model. The feedback loops in the dynamical model coincide with cycles in the directed graph $\mathcal{G}(\mathcal{B}_{\text{nat}}, M_{\text{nat}})$ that is obtained by applying step 1 of the causal ordering algorithm to the bipartite graph $\mathcal{B}_{\text{nat}} = \langle V_{\text{ext}}, F_{\text{ext}}, E_{\text{ext}} \cup E_{\text{nat}} \rangle$ using the perfect matching $M_{\text{nat}} = E_{\text{nat}}$.[7] The following theorem can be used to reason about the presence of partially unobserved feedback loops given a model and observations for a subsystem.

**Theorem 3.** *Consider a first-order dynamical model in canonical form for endogenous variables $V$ and an extension consisting of canonical first-order differential equations for additional endogenous variables $V_+$. Let $F$ and $F_{\text{ext}} = F \cup F_+$ be the equilibrium equations of the original and extended model respectively. Let $\mathcal{B} = \langle V, F, E \rangle$ be the bipartite graph associated with $F$ and $\mathcal{B}_{\text{ext}} = \langle V_{\text{ext}}, F_{\text{ext}}, E_{\text{ext}} \rangle$ the bipartite graph associated with $F_{\text{ext}}$. Assume that $\mathcal{B}$ and $\mathcal{B}_{\text{ext}}$ both have perfect matchings. If the model extension does not introduce a new feedback loop with the original dynamical model, then d-connections in $\mathrm{MO}(\mathcal{B})$ are also present in $\mathrm{MO}(\mathcal{B}_{\text{ext}})$.*

Theorem 3 characterizes a class of model extensions under which certain qualitative model predictions are robust, but it also shows how we can reason about the existence of unobserved feedback loops. To be more precise, it shows that, given a submodel for a subsystem, the presence of conditional independences that are not predicted by the submodel imply the existence of an unobserved feedback loop, if we assume faithfulness. If, for example, we assume that the viral infection model without an immune response is a submodel of the system that is described by the strictly positive equilibrium solutions of the viral infection model

---

[7]Note that a feedback loop in the dynamical model does not imply a feedback loop in the equilibrium equations as well. For example, there is feedback in the dynamical equations (3), (4), but there is no feedback in the causal ordering graph of the equilibrium equations in Figure 2b nor in the directed graph that is constructed in step 1 of the causal ordering algorithm.

with a single immune response, then we would observe an independence between $U_\sigma$ and $X_I$ that is not predicted by the model equations of the submodel. Theorem 3 would then imply that there is an unobserved feedback loop. Indeed, it can be seen from equations (3), (4), (7), (8) that there is an unobserved feedback loop from $X_I(t)$ to $X_E(t)$ to $X_\delta(t)$ and back to $X_I(t)$, while the Markov ordering graphs in Figures 2c and 3b imply that $U_\sigma$ and $X_I$ are dependent in the original model and independent in the extended model. We consider the use of existing structure learning algorithms for the detection of feedback loops in models with variables that are not self-regulating from a combination of background knowledge and observational equilibrium data to be an interesting topic for future work.

## 4.3 EXAMPLE: SIGNALING CASCADE MODEL

We will illustrate the ideas about detecting non-self-regulating variables and feedback loops by means of an example of a mathematical model for a dynamical system consisting of a signaling cascade of phosphorylated proteins. The model is a simplified version of that of [Shin et al., 2009], where we omitted the feedback mechanism through RAF Kinase Inhibitor Protein (RKIP) [Blom and Mooij, 2021]. The details of this model can be found in Section B of the Supplementary Material.

We denote the concentrations of active (phosphorylated) RAS, RAF, MEK, and ERK proteins, respectively, by $X_s(t)$, $X_r(t)$, $X_m(t)$, and $X_e(t)$, and denote by $I(t)$ an external stimulus or perturbation. The dynamics is modeled by differential equations (6*), (7*), (8*), and (10*) in Section B of the Supplementary Material. The full model consists of a signaling pathway that goes from $I(t)$ to $X_s(t)$ to $X_r(t)$ to $X_m(t)$ to $X_e(t)$, with negative feedback from $X_e(t)$ on $X_s(t)$. At equilibrium, we assume the exogenous input signal $I(t) = I$ to have a constant (possibly random) value, and let $f_s$, $f_r$, $f_m$, and $f_e$ represent the equilibrium equations (11*), (12*), (13*), and (14*) respectively.

Suppose now that the system is only partially modelled by treating the ERK concentration $X_e(t)$ as a latent exogenous variable in the submodel for the RAS, RAF and MEK concentrations ($X_s(t)$, $X_r(t)$, and $X_m(t)$, respectively) defined by equations (6*), (7*), and (8*). The complete model (including the differential equation (10*) that models the dynamics of ERK) can then be seen as a model extension of the submodel for RAS, RAF and MEK. Application of the causal ordering technique to the submodel (with $V = \{v_s, v_r, v_m\}$ and $F = \{f_s, f_r, f_m\}$) results in the Markov ordering graph in Figure 6a. Assuming faithfulness, the d-connections in this graph indicate that the equilibrium distributions for $X_s$, $X_r$ and $X_m$ all depend on the input signal $I$. Let us assume that we have observed data that is generated from the full model. The Markov ordering graph for the extended model (with $V^+ = \{v_e\}$, $F^+ = \{f_e\}$) is

displayed in Figure 6b, and implies that the equilibrium distributions for $X_s$, $X_r$ and $X_m$ are *independent* of the input signal $I$. Thus, what appears to be a faithfulness violation from the submodel perspective is explained by the Markov property of the extended model. In this case, the holistic modeling approach that allows for feedback through additional unobserved endogenous variables is needed, while the more common reductionistic assumption of treating all unobserved causes as exogenous to the observed variables will fail.

According to Corollary 1 and Theorem 3, the discrepancy between the observed and predicted conditional independences implies the presence of a non-selfregulating variable and an unobserved dynamical feedback loop (provided that we assume faithfulness). This is in agreement with the fact that the dynamic variable $X_e(t)$ is not self-regulating and that there is a feedback loop in the extended dynamical model. Remarkably, we can infer the presence of feedback in this way without explicitly modelling or even observing $X_e(t)$.

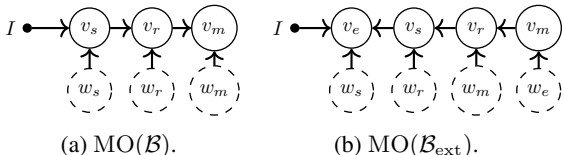

(a) MO($\mathcal{B}$).    (b) MO($\mathcal{B}_{\text{ext}}$).

Figure 6: Markov ordering graphs for the partial model (left) and the full model (right) of the RAS-RAF-MEK-ERK protein signaling cascade model.

## 5 DISCUSSION AND CONCLUSION

In this work we revisited several models of viral infections and immune responses. In our treatment of these models we closely followed the approach in De Boer [2012] and therefore we only considered strictly positive solutions. If we would have modelled all solutions then, for example, we would have considered the equilibrium equation $f_I : (U_f \beta X_T - U_\delta) X_I = 0$ instead of $f_I^+$ in equation (6). In that case, we would have obtained the causal ordering graph in Figure 7 instead of that in Figure 2b. Clearly, the model predictions of the causal ordering graph for the positive solutions in Figure 2b are more informative. The choice of only modelling strictly positive solutions depends on the application.

In many application domains mathematical models are used to predict the equilibrium behavior of complex systems. An important issue is that (causal and Markov) predictions may strongly depend on the specifics of the model design. We revisited an example of a viral infection model [De Boer, 2012], in which implied causal relations and conditional independences change dramatically when equations, describing immune reactions, are added. Analysis of this

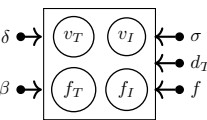

Figure 7: Causal ordering graph for positive and non-positive solutions of the viral infection model.

behavior through explicit calculations is neither insightful nor scalable. We showed how the technique of causal ordering can be used to efficiently analyze the robustness of implied causal effects and conditional independences under certain solvability assumptions. Using key insights provided by this approach we characterized large classes of model extensions under which predicted causal relations and conditional independences are robust. We hope that the results presented in this paper provide a step towards bringing the world of causal modeling and reasoning closer to practical applications.

Our results for the characterization of the robustness of model extensions can also be used to reason about the properties of models that are the combination of two submodels. This way, we can study systems whose causal and Markov properties can be understood in a reductionistic manner by considering the properties of its parts. When the properties of the whole model differ from those of its parts, a holistic modelling approach is required. For models of the equilibrium distribution of dynamical systems, we proved that extensions of dynamical models where each variable is self-regulating preserve the predicted presence of causal effects and d-connections in the original model. Based on those insights, we proposed a novel approach to model selection, where information about conditional independences can be used in combination with model equations to reason about possible model extensions or the presence of feedback mechanisms. For dynamical models with feedback, the output of structure learning algorithms does not always have a causal interpretation in terms of soft or perfect interventions for the equilibrium distribution. We have shown that in dynamical systems where each variable is self-regulating the identifiable directed edges in the learned graph do express causal relations between variables.

**Acknowledgements**

We thank Johannes Textor for introducing us to De Boer's paper and for interesting discussions about the viral infection model. We are also grateful to an anonymous reviewer for pointing out some flaws in an earlier version of this manuscript. This work was supported by the ERC under the European Union's Horizon 2020 research and innovation programme (grant agreement 639466).

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
