# OpenReview forum: "Robustness of Model Predictions under Extension"
_auai.org/UAI/2022/Conference — UAI 2022 Oral_

### Official Review · Reviewer_YKsd · 2022-04-10

**Q2(1) Originality/Novelty:** 3
**Q2(2) Significance/Impact:** 2
**Q2(3) Correctness/Technical Quality:** 3
**Q2(6) Clarity Of Writing:** 3
**Q6 Overall Score:** 6
**Q8 Confidence In Your Score:** 3

**Q1 Summary And Contributions:**

This paper considers the stability of a model's predictions about the presence and absence of causal effects and conditional independencies upon extending the model with additional variables and equations. While in general such predictions are shown to be not stable under model extension, using a graphical framework the authors present sufficient conditions under which they are. The theoretical results are illustrated with the running example of a model for immune system responses.

**Q10 Ethical Concerns (Optional):**

No ethical concerns.

**Q2 Assessment Of The Paper:**

More detailed information regarding each of these aspects is given below:

**Q2(5) Reproducibility:**

3: Good: Key resources (e.g., proofs, code, data) are available and key details (e.g., proofs, experimental setup) are sufficiently well-described for competent researchers to confidently reproduce the main results.

**Q3 Main Strengths:**

The paper is clearly written and very precise in most of its statements. The idea of using a graphical framework to reason about the stability of predictions about the presence and absence of causal effects and conditional independencies and, by turning this around, to reason about admissible model extensions is interesting. I think this can be useful for modellers. The theoretical results appear to be technically solid, although I did not check their proofs in the appendix.

**Q4 Main Weakness:**

I am slightly concerned that some of the theoretical results are not-so-surprising findings packaged in a rather intricate graphical terminology, see also my comments on Theorems 1 and 2 in Q5. But I cannot say this with confidence and, anyway, the graphical formalization can still be of value. The possible connection to causal discovery that is advocated at the end of section 4 remains unclear to me. For both these two reasons I am not sure how large the paper's impact will be.

**Q5 Detailed Comments To The Authors:**

Major comments:

- "Since there is no directed path from $v_{\delta}$ to $v_I$ in the Markov ordering graph, the effect of this perfect intervention would not have been predicted by the Markov ordering graph, if it would have been interpreted causally. Therefore, contrary to the causal ordering graph, the Markov ordering graph does not have a causal interpretation ...": This confuses me. Also in the causal ordering graph there is no directed path $v_{\delta}$ to $v_I$ (and also not from $f_{\delta}$ to $v_I$), so wouldn't this mean that also the causal ordering graph does not have a causal interpretation in terms of perfect interventions?

- Theorem 1: Are you aware of conditions under which $\mathcal{B}+$ has perfect matching? If yes, this would greatly aid the intuition. For example, is it sufficient that none of the equations in $F_+$ only involves variables from $V$? This is violated by the single immune response extension, see equation (9), which is the reason why a) in $\mathcal{B}_+$ there is no perfect matching and why b) there is an additional constraint on the variables in the subsystem. So at least in this example the same conclusion can be anticipated already by looking at equation (9).

- Theorem 2: The condition "no vertex in $V_+$ is adjacent to a vertex in $F$ in $\mathcal{B}{\text{ext}}$" in conjunction with footnote 6 can be rephrased to saying that no variable in $V_+$ appears in an equation in $F$, right? If yes, then to me this seems like a much easier way of stating the assumption. Again if yes, then it seems to me the statement becomes almost intuitively clear because the subsystem added by the model extension does then by construction not have an effect on the original subsystem.

- Corollary 1 and the subsequent discussion: Perhaps I am wrong, but I think the discussion is not entirely adequate. Namely: If all variables in the extended model are self-regulating, then also all variables in the original model and all variables in the extension part are self-regulating, right? This is why under the assumptions of Corollary 1 both $\mathcal{B}$ and $\mathcal{B}_+$ have a perfect matching and Theorem 1 applies. However, considering the viral infection model example, already the model without immune response is not self-regulating, see equation (6). So in this example Corollary 1 cannot be used to draw any non-trivial conclusion about the extended model---whereas the discussion in section 4.1 seems to reason otherwise. (Yes, Corollary 1 does say that the extended system is not self-regulating, but this is already trivially implied because already the subsystem without immune response is not self-regulating.) Am I missing something?


Minor comments:

- "Can be used to better understand the qualitative model predictions of these systems": Please be already here specific about what type of predictions you are referring to.

- Section 1.1: Here you seem to assume the existence of a perfect matching. Does the paper say anything about cases in which there is no perfect matching? This would be helpful to clarify in the final version.

- "where $\mathcal{V}$ is a partition of vertices $V$ into clusters" and "Let $\mathcal{V}$ be the set of clusters $S_i \cup M(S_i)$: There seems to be a slight mismatch here. According to the former statement $\mathcal{V}$ consists of sets of vertices in $V$, according to the second statement of sets of vertices in $V \cup F$.

- "that are matched to vertices in $S_i$ in matching $M$": You could improve readability by specifying what this means.

- "for a more elaborate cyclic model can be found in Appendix A.2": I think it is Appendix A.1.

- "It can be shown that a) the presence ...": Is there a similar statement about perfect interventions? In either case, this would be useful to state.

- The part of section 4 on page 6: This discussion is unclear to me. Rejecting the model extensions would require a) the assumption that the equations for the original subsystem are correct and b) that the (conditional) independencies are measured/tested. Please state this explicitly, respectively clarify the discussion.

- "if it can be solved uniquely from the equilibrium equation that is constructed from its derivative": Solved in terms of what? Please be more precise here.

- Footnote 7: I think this an important statement that deserves more than being a footnote.

**Q7 Justification For Your Score:**

The strengths outweigh the weaknesses. The paper appears to be technically solid and contains potentially useful results.

**Q9 Complying With Reviewing Instructions:**

1: Yes.

---

### Official Review · Reviewer_7f5M · 2022-04-12

**Q2(1) Originality/Novelty:** 2
**Q2(2) Significance/Impact:** 2
**Q2(3) Correctness/Technical Quality:** 3
**Q2(6) Clarity Of Writing:** 3
**Q6 Overall Score:** 5
**Q8 Confidence In Your Score:** 2

**Q1 Summary And Contributions:**

The paper discussed about the observation of combination model predictions of two systems and showed the causal ordering technique to assess the robustness of qualitative model predictions. The author characterized a large class of model extensions that preserve qualitative model predictions and demonstrated how causal ordering can be used as a scalable tool to assess the robustness of model predictions without requiring explicit calculations.

**Q2 Assessment Of The Paper:**

More detailed information regarding each of these aspects is given below:

**Q2(4) Quality Of Experiments (Optional):**

3: Good: The experimental evaluation is adequate, and the results convincingly support the main claims.

**Q2(5) Reproducibility:**

3: Good: Key resources (e.g., proofs, code, data) are available and key details (e.g., proofs, experimental setup) are sufficiently well-described for competent researchers to confidently reproduce the main results.

**Q3 Main Strengths:**

- The author provided equation and figure to support the explanation.
- The author demonstrated the way to gauge the robustness of model predictions with the example.
- The author demonstrated how causal ordering can be used as a scalable tool to assess the robustness of model predictions without requiring explicit calculations

**Q4 Main Weakness:**

- the author proposed existing technique to asses the robustness of qualitative causal predictions.
- the author should provide more explanation on how/why they make a conclusion
- the author should provide a profound explanation of the novel insights that they propose


**Q5 Detailed Comments To The Authors:**

What are the examples of the model prediction that is not robust? What are the vulnerabilities of these models?

Why the author applies the ideas to a viral infection model with immune responses? Is there any specific reason?

Page 4:
The author needs to provide more explanation how they concluded that the independency between Uσ and XT that was implied by the Markov ordering graph is not robust?

**Q7 Justification For Your Score:**

- the author provided good explanation of equations and figures
- the author provided proof in appendix to reproducibility

**Q9 Complying With Reviewing Instructions:**

1: Yes.

---

### Official Review · Reviewer_39re · 2022-04-12

**Q2(1) Originality/Novelty:** 3
**Q2(2) Significance/Impact:** 2
**Q2(3) Correctness/Technical Quality:** 3
**Q2(6) Clarity Of Writing:** 4
**Q6 Overall Score:** 5
**Q8 Confidence In Your Score:** 3

**Q1 Summary And Contributions:**

A causal ordering model represents the causal relations between variables in equations in a graph with clusters, nodes and directed edges.
The authors investigate how the model changes when it is extended by additional equations, and prove conditions under which the relations in the model do not change when extending.

**Q2 Assessment Of The Paper:**

More detailed information regarding each of these aspects is given below:

**Q2(5) Reproducibility:**

3: Good: Key resources (e.g., proofs, code, data) are available and key details (e.g., proofs, experimental setup) are sufficiently well-described for competent researchers to confidently reproduce the main results.

**Q3 Main Strengths:**

The research is practically motivated with the  example of a viral infection model.

They prove when such a causal ordering model does not change from extensions.

**Q4 Main Weakness:**


A large part of the paper is dedicated to explaining the viral infection model, although that does not matter much for artificial intelligence.


I have never seen a causal model like this, so I doubt the results will have much impact.

**Q5 Detailed Comments To The Authors:**


>p2 where (cal) V is a partition of vertices V into clusters

vertices V  and vertices F ? The example clusters V and F

>p4 The causal ordering graph tells us that soft interventions targeting Uσ , Uf , Uδ , dT , or β generically have an effect on the equilibrium distribution of the amount of infected cells XI

Uσ , Uf , Uδ , dT , or β are all variables in the model? So targeting any variable in the model has such an effect?


>p5 By explicitly solving equilibrium equations it is easy to check that replacing fδ with an equation setting Xδ equal to a constant generically changes the distribution of XI .

When the equations are valid in the equilibrium, and an intervention  changes the equilibrium, are the equations and the model still valid?

> Theorem  1 If B and B+ both have a perfect matching then

Is that something that happens often in practice?

>p7 Corollar 1 If all variables in V ∪ V+ are self-regulating then 2 and 3 of Theorem 1 hold.

1 of Theorem 1 also holds? Why not mention it? Also ", then"



**Q7 Justification For Your Score:**

Extending causal models can be important for reasoning about uncertainty. The viral infection model did not influence my score.

**Q9 Complying With Reviewing Instructions:**

1: Yes.

---

### Official Review · Reviewer_hChq · 2022-04-16

**Q2(1) Originality/Novelty:** 3
**Q2(2) Significance/Impact:** 3
**Q2(3) Correctness/Technical Quality:** 3
**Q2(6) Clarity Of Writing:** 4
**Q6 Overall Score:** 7
**Q8 Confidence In Your Score:** 3

**Q1 Summary And Contributions:**

This manuscript provides theoretical results and examples regarding the preservation of causal effects and conditional independences under model extensions.

**Q2 Assessment Of The Paper:**

More detailed information regarding each of these aspects is given below:

**Q2(4) Quality Of Experiments (Optional):**

3: Good: The experimental evaluation is adequate, and the results convincingly support the main claims.

**Q2(5) Reproducibility:**

4: Excellent: Key resources (e.g., proofs, code, data) are available and key details (e.g., proof sketches, experimental setup) are comprehensively described for competent researchers to confidently and easily reproduce the main results.

**Q3 Main Strengths:**

- the paper is well written,
- the question addresses is very important,
- illustrations with viral infections model are very insightful,
- section 4 provides very nice hints on how to exploit the framework to draw conclusions from observational data.

**Q4 Main Weakness:**

There is no experimental results to illustrate the framework

**Q5 Detailed Comments To The Authors:**

While the generic effect of interventions can be read from the causal ordering graph, I am left wondering whether interventional/multi-environment data  (e.g. the outcome of scientific experiments) can be further exploited to reason about/select extensions.

**Q7 Justification For Your Score:**

The authors address a conceptually hard problem with a lot of effort on providing the reader with insights and examples. The theoretical results are perhaps not too hard to get based on previous work on Markov and Causal ordering graphs, but they are definitively useful.

**Q9 Complying With Reviewing Instructions:**

1: Yes.

---

### Decision · Program_Chairs · 2022-05-15

**Decision:**

Accept (Oral)

**Comment:**

Meta Review: An interesting paper that examines the extent to which causal models can be analyzed as modular systems. It is well known that in the presence of feedback coupling together different systems can lead to a system that collectively behaves very differently from the components studied individually. For example systems that are unstable individually can become stable when joined together.
This paper examines under which circumstances will the behavior of a system be preserved when the model is "extended" by adding it to a larger system.

There is some difference in opinion among the reviewers regarding significance and impact, but this seems to derive in part from different views over the importance of feedback models to AI. Though such models are not currently very common this can certainly be seen as a "feature" rather than a "bug" since it shows that the paper is original and extending techniques to qualitatively new systems. Certainly regardless of whether they are common in AI, feedback and homeostasis is central to all biological systems!

The consensus of the reviewers is that this is a good paper that is generally well-written and should be accepted.